# Efficiency of Digital Photolithographic Synthesis of Large, High-Quality DNA Libraries and Microarrays using a Guanine O⁶ Dephosphitylation Strategy

Santra Santhosh [1,2], Sharon Istvánffy [2,3], Omer Sabary [4], Eitan Yaakobi[4], Maya Giridhar[2], Jürgen Behr [2] & Mark M. Somoza [2,5,6] ✉

Large-scale de novo nucleic acid synthesis is a powerful tool enabling researchers to better understand and engineer biological systems. Fields ranging from genomics to nucleic acid therapeutics to synthetic biology make use of high-throughput experimental approaches requiring access to large pools or libraries of DNA, RNA, synthetic nucleic acid analogs, non-nucleosidic building blocks, or combinations of these. Large oligonucleotide libraries are synthesized as microarrays and used in situ for surface-based assays or cleaved for off-array applications. Here, using a digital maskless photolithographic approach, we address an important source of error in DNA microarray synthesis, oligonucleotide fragmentation arising from the O⁶-phosphitylation of guanine during the potentially hundreds of coupling cycles required for complex library synthesis. Introducing a very short debranching step using standard capping reagents suppresses depurination-based fragmentation and greatly enhances synthetic yield.

Nucleic acid microarray synthesis technology is the only source of very large, designed-sequence oligonucleotide libraries[1]. These libraries have become essential for a wide range of modern research fields, including genomics, synthetic biology, nanobiotechnology, and nucleic acid therapeutics. An emerging application is in information technology, where recent years have seen a strong effort to use DNA as a carrier of digital data for archival storage[2–8] or for embedding information into everyday products ("the DNA of things"[9]), both for labeling and for authentication[10,11]. The original approach to microarray synthesis was the photomask-based photolithographic technique introduced by Affymetrix, which has been used commercially for about 30 years for high-throughput genomics applications[12–14]. This approach was refined by digital maskless nucleic acid photolithography, which replaced the physical chrome masks with digital micromirror device (DMD) technology[15], enabling its implementation as a tabletop device suitable for routine laboratory use[16–19]. Digital maskless nucleic acid photolithography is unique in that its chemistry has been significantly optimized and extended to allow synthesis also with RNA[20–23],

L-DNA[24], 2′-fluoroarabinonucleic acid (2′F-ANA)[25], 2′OMe-RNA[26], non-nucleosidic building blocks[27], peptides[28], peptide nucleic acid (PNA)[29], and fluorescent labels[30,31].

Two well-established alternative approaches to photolithographic microarray synthesis of DNA are: (1) inkjet printing, replacing the color inks with activated phosphoramidites[32–35], and (2) digital spatial control of deblocking (detritylation) using semiconductor-microchip-based arrays of individually addressable microelectrodes that generate a localized acidic environment[36]. Despite their unique features, all methods use modified forms of phosphoramidite chemistry[37] and are affected by many of the same chemical side reactions during the synthesis of long oligonucleotides. The primary chemical difference among microarray synthesis approaches is the deblocking strategy. Both ink-jet printing and electrochemical synthesis employ acid-mediated removal of the dimethoxytrityl (DMTr) 5′-hydroxyl protecting group[38]. Acidic conditions during deblocking promote depurination, which leads to the formation of abasic sites that cause strand cleavage during final deprotection[35,39]. Nucleic acid photolithography

¹Technical University of Munich, TUM School of Natural Sciences, Garching, Germany. ²Leibniz Institute for Food Systems Biology at the Technical University of Munich, Lise-Meitner-Straße 30, 85354 Freising, Germany. ³Technical University of Munich, TUM School of Life Sciences, Freising, Germany. ⁴The Henry and Marilyn Taub Faculty of Computer Science, Technion, 3200003 Haifa, Israel. ⁵Chair of Food Chemistry and Molecular Sensory Science, School of Life Sciences, Technical University of Munich, Lise-Meitner-Straße 34, 85354 Freising, Germany. ⁶Department of Inorganic Chemistry, Faculty of Chemistry, University of Vienna, Josef-Holaubek-Platz 2, 1090 Vienna, Austria. ✉e-mail: mark.somoza@univie.ac.at

replaces the DMTr group with a photolabile group, originally α-methyl-2-nitropiperonyloxycarbonyl (MeNPOC)[40], and more recently, 2-(2-nitrophenyl)propoxycarbonyl (NPPOC), or a derivative thereof, either benzoyl-2-(2-nitrophenyl)propoxycarbonyl (Bz-NPPOC) or thiophenyl-2-(2-nitrophenyl)propoxycarbonyl (SPh-NPPOC)[41], all of which eliminate exposure to the strongly acidic conditions that result in depurination. Therefore, the use of photolabile groups may allow the synthesis of longer oligonucleotides, although photochemical side reactions may constrain the length in ways that are not yet fully elucidated. The use of MeNPOC-protected DNA phosphoramidites results in deficient coupling due to the incomplete recovery of 5′ hydroxyl groups after photolysis, due to capping by its photobyproducts[14]. NPPOC derivatives release a much less reactive α-methylstyrene. While they can form problematic nitroso derivatives during photolysis, these are eliminated with the use of basic photodeprotection conditions via non-nucleophilic heterocyclic secondary or tertiary amines, such as 1% imidazole in DMSO, as an exposure solvent during light irradiation of the synthesis surface[42–45].

Beyond deblocking, the coupling and oxidation chemistry of microarray synthesis is very similar to that of standard solid-phase synthesis. Phosphoramidite coupling is generally more efficient in microarray synthesis due to the very low synthesis scale of about 20 pmol per square centimeter of array surface, allowing a vast excess of the coupling reagents[20,46–48]. Also, in photolithographic synthesis, the substitution of light for the standard acidic deblocking step allows for a reduced oxidation frequency, but only when DCI activator[49] is used, due to its low acidity (p$K_a$ 5.2), which is insufficiently acidic to cleave the phosphite triester formed during coupling reactions[47,50]. This absence of acid-mediated depurination suggests that photolithographic synthesis is able to achieve longer oligonucleotide lengths. Sequence lengths in this project reached 105mers, and we routinely synthesize complex arrays of 120mers; we are exploring library synthesis of 150mers and above.

Capping strategy is also a potentially important difference. In standard solid-phase synthesis, capping is carried out after the coupling reaction by acetylation of unreacted 5′-hydroxyls. In microarray synthesis, capping is considered unnecessary since the major impurities, sequences with deletions due to failed coupling reactions or deblocking, cannot be selectively removed from the surface. And even if the sequences are to be cleaved from the surface, the complexity of the resulting library, potentially hundreds of thousands up to a few million unique sequences, combined with the very small synthesis scale, makes purification difficult. To the extent that on-array capping is sometimes useful[17,51,52], and because acetylation is relatively inefficient and slow and can adversely modify surface characteristics, phosphoramidite capping reagents have been developed for microarray use[51,52]. Diethylene glycol ethyl ether ("UniCap") phosphoramidite is available commercially. Even standard DMTr phosphoramidites can be used for efficient capping in photolithographic synthesis, since, without an acid deblocking step, the DMTr remains attached throughout the synthesis and blocks further coupling reactions[53]. Phosphoramidite capping agents are extremely fast and effective compared to acetylation[46,52,54]. However, Pon et al observed that in standard solid-phase synthesis of oligonucleotides, activated phosphoramidites could phosphitylate the lactam function of previously coupled guanosines[55]. They showed that this reaction is followed by phosphorous group migration to $N^7$, resulting in a depurination-prone $N^7$-modified guanosine, and ultimately, strand cleavage during final deprotection. This chain cleavage could be mitigated by using 2-cyanoethyl or *p*-nitrophenylethyl protecting groups at the $O^6$ position of guanine. The former requires a more complex synthesis, while the latter is a photolabile protecting group incompatible with photolithographic synthesis. A proposed alternative using only the standard protecting group for the exocyclic amine of guanine was to regenerate the intact nucleobase immediately after the coupling reaction via a nucleophilic exchange reaction with the acetate ions present in the capping reagent[56–58]. Here, we demonstrate that the digital maskless photolithographic synthesis of DNA is prone to unexpectedly extensive phosphitylation of guanine, resulting in branching followed by depurination and cleavage, likely as a result of the very large

number of coupling reactions required for complex library synthesis. After quantifying the extent of these problems, we show that by adapting the results of Pon et al, this important side reaction can be effectively and efficiently eliminated using capping reagents. Unlike the acetylation reaction for capping, the debranching reaction using the same reagents is complete within a few seconds and results in a much higher yield of intact PCR amplifiable and sequenceable DNA strands and greatly improved fluorescence signals in hybridization-based assays.

## Results and discussion

We used digital maskless photolithographic synthesis of DNA microarrays and microarray-derived libraries to determine the extent to which phosphitylation of guanine affects oligonucleotide integrity, as well as to optimize the use of capping reagents to minimize branching and fragmentation while minimizing the extension of synthesis time. Figure 1a–c illustrates the chemistry and products created by the $O^6$ phosphitylation of guanine during the synthesis. The "Normal" cycle (N) omits the exposure to acetate ions, which results in branching. Within this cycle, the branching side-product chemistry is shown as the "Branching" cycle (B). Adding the acetate ion exposure between the coupling and oxidation steps results in the "Debranching" cycle (D). The diversity of fragmentation products in the case of the synthesis of a simple hypothetical DNA 6-mer with a single dG at the 4th position is shown in Fig. 1d. There are three product types: the array-bound fragment preceding the dG (which is also released in the case of library synthesis), the variable length 3′ guanine branches resulting directly from guanosine depurination, and the 5′ fragment of the designed oligonucleotide, a 3′ α,β-unsaturated aldehyde product resulting from base-catalyzed cleavage of the phosphodiester backbone 3′ to the abasic site. Figure 1d also shows guanine phosphitylation by another guanosine phosphoramidite during the same coupling reaction. Since the coupling reaction with a large excess of incoming phosphoramidite is both very fast and follows pseudo-first-order kinetics, the initial dG coupling can likely be followed by a branching event with a dG phosphoramidite from the same coupling cycle with a probability similar to that of subsequent coupling cycles.

### Role of guanine

We hypothesized that fragmentation could be due to depurination[39], particularly since very large numbers of coupling cycles are typical for microarray synthesis, typically 3–4 times oligonucleotide length, potentially as many as several hundred, depending on sequence library length and complexity.

We compared the synthesis of a mixed-base 61mer sequence (GTT AAG CGA AGA AGA AAG TAG CGT GGC GCA CAG TTG CCC AAT CAA TTA CAC CCT CAT TTC T) with the same sequence with all Gs replaced with Ts (TTT AAT CTA ATA ATA AAT TAT CTT TTC TCA CAT TTT CCC AAT CAA TTA CAC CCT CAT TTC T), using different activators with decreasing p$K_a$, to see if more acidic activators (BTT, p$K_a$ = 4.08 < ETT, p$K_a$ = 4.28 < DCI, p$K_a$ = 5.2) increase fragmentation. Nucleobase-specific fragmentation cannot be trivially studied using gel electrophoresis analysis of homopolymers or sequences with mixed patterns of purines and pyrimidines due to nucleobase-specific labeling biases of nucleic acid fluorescence stains[59]. Gel Red, an efficient stain for single-stranded DNA, has a much stronger binding affinity with thymine, therefore requiring that quantitative comparisons only be made between T-rich sequences with similar base compositions[60]. A comparison of the same mixed-base 61mer sequence synthesized using different coupling activators showed fragmentation levels were similar, demonstrating the absence of depurination due to acidic activators (Fig. 2; Supplementary Fig. 3). The sequences used in these experiments are shown in Supplementary Table 7. The same 61mer sequence, but with Ts replacing all Gs, exhibits greatly reduced fragmentation. Similar experiments with adenine replacing guanine in an otherwise T homopolymer sequence indicated that the side reaction does not occur at A positions (see Supplementary Fig. 4). This demonstrates that the fragmentation is due to G-specific side reactions.

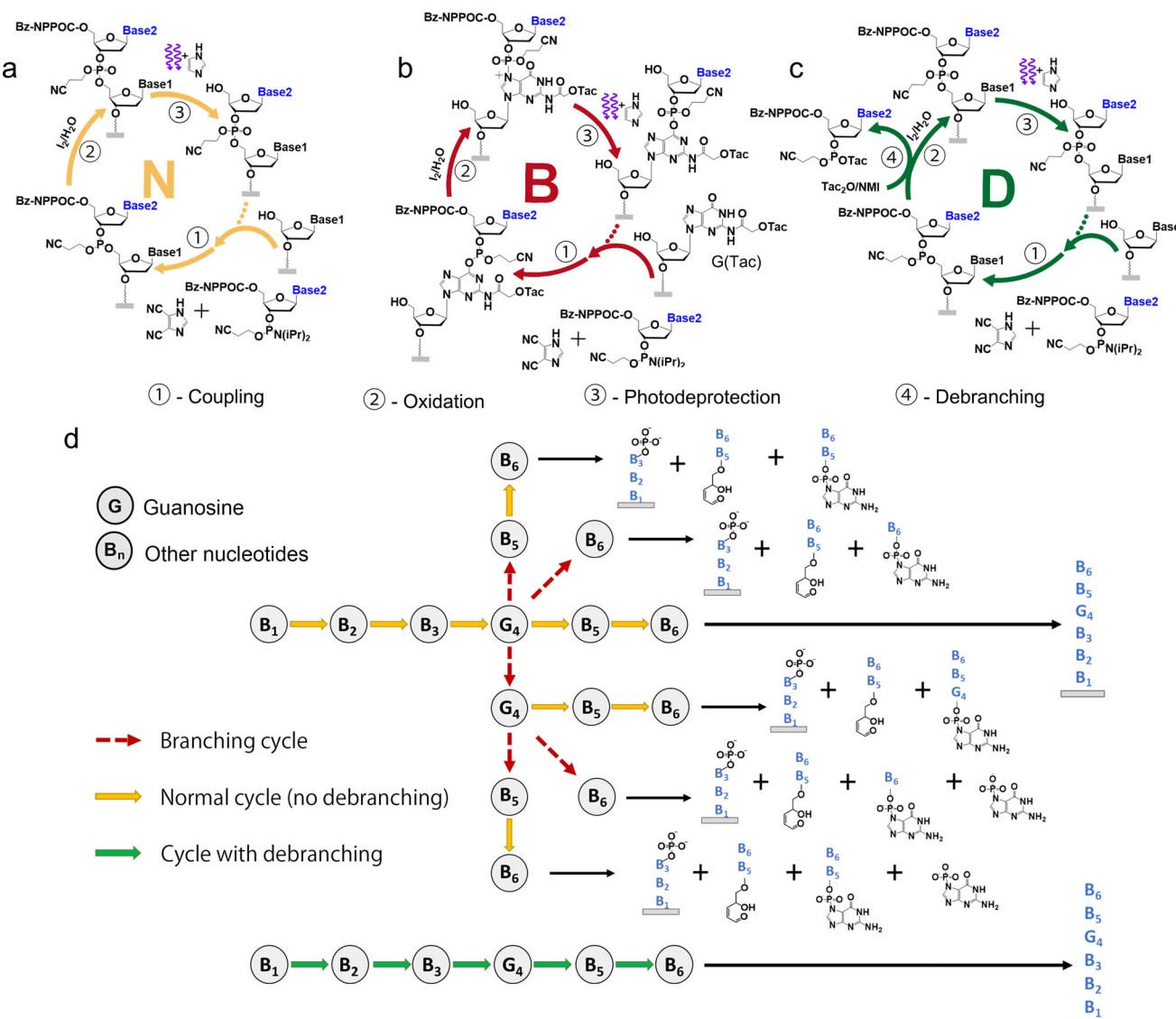

① - Coupling    ② - Oxidation    ③ - Photodeprotection    ④ - Debranching

**Fig. 1 | Schematics on the chemistry and products created by the O⁶ phosphity-lation of guanine during the digital photolithographic synthesis of DNA.**
**a** Chemical cycle of "Normal" photolithographic synthesis without a debranching step. **b** Guanine nucleobases previously coupled during the normal synthesis can undergo O⁶ phosphitylation, followed by migration of the phosphorous group to the N⁷ atom, as shown in the "Branching" cycle. This "Branching" cycle is a side reaction of the "Normal" cycle. **c** Extending the "Normal" cycle with a brief exposure to acetate ions recovers the intact guanine nucleobase, as shown in the "Debranching"

cycle. **d** Branching products, released during final chemical deprotection, from the synthesis of a DNA 6-mer with a single dG at position 4. Each coupling reaction following the dG coupling, including the dG coupling itself, potentially results in O⁶ phosphitylation, followed by migration to the N⁷ position, then elongation of the resulting branch during subsequent cycles. The fragments are the array-bound 3mer, the 3′ guanine depurination branch products, and the 3′ α,β-unsaturated aldehyde product resulting from base-catalyzed cleavage of the phosphodiester backbone 3′ to the abasic site.

## Effect of guanine on chain cleavage

Guanine-specific fragmentation was analyzed by introducing a deb-ranching step, using capping reagents, directly post-coupling. Here, we will use the term "debranching" to refer to the use of capping to reverse the guanine modification. Some degree of actual capping is presumably also achieved, but particularly for very short debranching times, the capping effect is weak, also due to the very high coupling efficiencies achieved, and therefore the primary effect is considered to be debranching. A $T_{60}$ homopolymer with two G substitutions at coupling cycles 20 and 21 (TTT TTT TTT TTT TTT TTT TTT TTT TTT TTT TTT TTT GGT TTT TTT TTT TTT TTT TTT) was synthesized without debranching and with debranching times from 0 s (flow through with no waiting time) up to 60 s. Representative protocols for synthesis with and without debranching can be found in Supplementary Table 1. The debranching reagent was a standard capping solution, an equimolar mix of tert-butylphenoxyacetyl

acetic anhydride ($Tac_2O$) in tetrahydrofuran and 10% N-methylimidazole in tetrahydrofuran/pyridine (80:10), and was able to mostly eliminate fragmentation at all exposure times, while omission of the debranching (-) results in a clear pattern of fragmentation corresponding to the position of the G pair (Fig. 2b). A 15 s debranching treatment on three types of sequences (Fig. 2c) also mitigated fragmentation effectively vs a control without the debranching reagent. The three sequences were: (1) the mixed-base 61mer sequence (lanes one and two), (2) the $T_{60}$ homo-polymer with G substitutions at positions 20 and 21 (lanes 3 and 4), and (3) as well as a complex library of around 80,000 distinct oligonucleotide 61mers (corresponding to probes from a human gene expression microarray[61]). In the case of the 45,000 61mer library with debranching, the product band (lane 6) extends both above and below the 60mer ladder position due to the heterogeneity of the library, resulting in interactions between sequences.

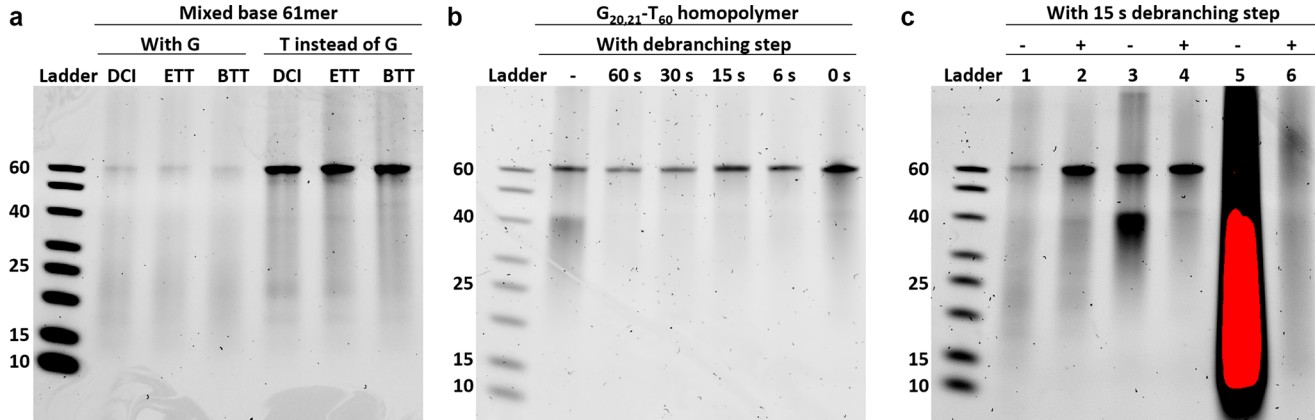

**Fig. 2 | Investigation on guanosine-specific fragmentation of oligonucleotides cleaved from the microarrays (15% TBE-urea PAGE analysis). a** Mixed-base 61mer sequences, either with G or with Ts replacing all Gs, with coupling activators with different acid strengths, DCI, ETT, and BTT, to verify guanine-dependent strand cleavage. **b** Verification of $O^6$-phosphitylation of guanine and its regeneration with exposure to capping/debranching reagents for different times up to 60 s. The synthesized oligonucleotide is a 60 nt T homopolymer with two G substitutions at positions 20 and 21. The symbol (−) indicates no debranching step, while 0 s indicates that the debranching reagents were introduced but immediately washed away. **c** Verification of the effect of applying a 15 s debranching step (+) or not (−) on three types of sequences: mixed-base 61mer (lanes 1 and 2), $T_{60}$ homopolymer with G substitutions at positions 20 and 21 (lanes 3 and 4), and a complex array of circa 80,000 different mixed-base 61mers (lanes 5 and 6). The red color in lane 5 indicates saturation due to overexposure required to make all lanes visible (see Supplementary Fig. S2 for an unsaturated version). All the gel images used the same fraction of total array-synthesized DNA.

## Hybridization assay

We hypothesized that the number and position of Gs in sequences would influence oligonucleotide integrity in the absence of a debranching reaction, since G couplings closer to the 3′ end would be exposed to more coupling cycles and hence more likely to undergo depurination (3′ to 5′ synthesis was used exclusively). This effect can be expected to be more pronounced in microarray synthesis vs standard solid phase synthesis since several coupling cycles are required on average for all the oligonucleotides on the surface to be extended by a single nucleotide (see Supplementary Table 8 for the number of synthesis cycles in each design).

To investigate the effect of one or more G positions and the number of Gs at each position on strand integrity using a hybridization intensity assay, two microarray sets, one with and the other without a debranching step in each cycle, were synthesized. The fluorescence intensities for each array were normalized to that of the pyrimidine 40mer devoid of guanines. All oligonucleotides were synthesized on a $T_5$ linker and ended with a 25mer reporter probe (Fig. 3a). The sequences used in the hybridization assay are given in Supplementary Table 2. With the use of debranching, the hybridization intensities remained relatively consistent throughout sequences with different G-rich positions, with a gradual decrease from "GG" to "GGG" to "GGGG". Conversely, in the absence of debranching, moving the guanosine content to the 3′ end resulted in a progressive decline in intensity, particularly pronounced for "GGGG" (Fig. 3b, c). Without debranching, G-rich sequences containing 12 Gs within the otherwise pyrimidine-only 40mer resulted in a very weak hybridization signal (Fig. 3d).

## Gene expression array analysis

Gene expression microarray synthesis is a good stress test of large-scale DNA synthesis because such arrays contain a very large number of unique 60mer sequences. Our human gene expression array contains a minimum of three independent sequence probes for each of more than 45,000 human genes and gene variants, for a total of more than 135,000 unique 60mer probes, and requires 160 synthesis cycles. The average G content of the transcript probes is 24%, somewhat lower than the 30% of the G-rich sequence in the hybridization assay above. However, the gene expression array synthesis requires far more coupling reactions; therefore, similar, or perhaps more extensive, degradation was expected in the absence of a debranching step. In agreement with this expectation, the gene expression microarrays synthesized with a 15 s debranching step have a greatly enhanced hybridization performance. In particular, a sixfold increase in fluorescence signal intensity and a significant increase in signal/noise highlight the effectiveness of debranching. This is illustrated in Fig. 4a, c, which show close-ups (approximately 0.4% of the total area) of the fluorescence images of arrays synthesized without and with, respectively, the debranching step. The significantly improved hybridization signal results in a lower signal-to-noise ratio for the resulting transcription data. This can be seen in Figs. 4b, 4d, which show scatterplots for the gene expression arrays synthesized without and with debranching, respectively.

## Quantitative PCR and sequencing analysis

Particularly for off-array applications of sequence pools or libraries, hybridization-based assays are likely insufficient to judge oligonucleotide yield and quality. To address debranching-dependent cleaved library quality, we synthesized arrays using a base-cleavable linker to generate oligonucleotide libraries for analysis using qPCR as well as Illumina sequencing. The library design is similar to that used for the hybridization assays (Fig. 3a), with the number and positions of G-rich sections at various positions within an otherwise pyrimidine-only test sequence (Fig. 5a, Supplementary Table 3). Specifically, a 53mer pyrimidine-only test sequence was used as a control, and G triplet substitutions at increasing distances from the 3′-end were introduced, as well as a sequence with three sets of G triplets. The resulting sequence libraries were cleaved from microarrays synthesized with and without debranching and amplified. Five ng/µl of each sample, as measured by Nanodrop spectrophotometry, were used. The quantitative PCR (qPCR) amplification curve of the library synthesized without debranching is shifted by $\Delta C_q = 7.84$ cycles, indicating a much lower number of amplifiable sequences relative to the library synthesized with debranching (Fig. 5b).

Since the sample from the array synthesized without debranching includes a large fraction of sequence fragments, the absorption-based quantitation is biased due to the hypochromism of polynucleotides, which results in a decreasing absorption with increasing polymerization[62]. Since the fragmentation results in a decrease in polymerization, the absorption measurement will overestimate the amount of DNA present in the sample. In the absence of a debranching step, the branches themselves additionally contribute to the amount of DNA detected, resulting in an approximate seven-fold increase in the measured concentration vs. the same library synthesis with a debranching step (Supplementary Table 4). Therefore, the qPCR quantitation provides a reasonable estimate of the relative yield because fragments and branches will not amplify. Comparing Nanodrop

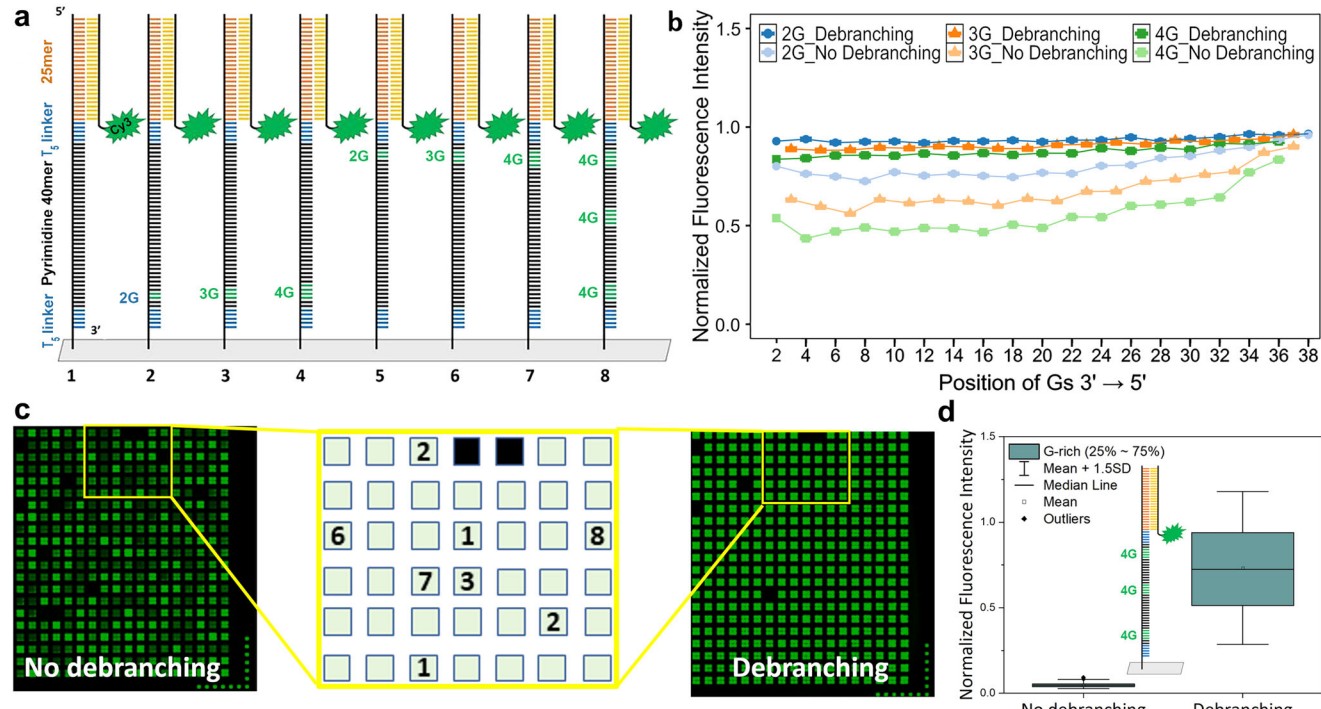

**Fig. 3 | The effect of position-dependent G-rich centers on hybridization intensity with and without debranching. a** Schematic representation of the sequence design. Each sequence has a $T_5$-linker, a 40mer pyrimidine sequence $(CTCT)_n$ test probe. These test probes were inserted with either no G-rich positions or "GG", "GGG", "GGGG" (represented as 2 G, 3 G, and 4 G, respectively, in the figure) at different positions from the 3' to the 5' end, one set of G insertions per sequence. This test probe was then followed by a $T_5$-linker again and a 25mer reporter probe at the 5' end, which was then hybridized with 5'-Cy3-labeled complementary 25mer. **b** Fluorescence intensity decreases rapidly for the sequences with G-rich positions at the 3' end in the absence of debranching. The signal intensities were normalized to the corresponding fluorescence signal of the pyrimidine

sequence without G insertions as a control and plotted versus the position of the G incorporations within the 40mer pyrimidine sequence. Each point is the average of 1959 replicates on one array, and the error bars are SEM. **c** Fluorescent scan image of arrays synthesized with and without a 15 s debranching step (images show ~0.5% of the total area of each array). In the legend, black features denote background probes (no synthesis), and the numbered squares indicate the positions of some of the representative sequences shown in (**a**). The arrays were scanned with identical scanner settings. **d** Boxplot of normalized fluorescence intensity of a G-rich sequence (marked as 8 in the diagram in (**a**). Each boxplot is based on 1959 replicates of a G-rich sequence per array and two replicate arrays for each synthesis condition.

and qPCR concentration data indicates that libraries synthesized with debranching result in good agreement between the two measurements. For samples synthesized without debranching, Nanodrop overestimates the concentration by greater than a factor of 10 vs. qPCR (Supplementary Table 6), indicating a high degree of fragmentation consistent with the gel images of mixed-base sequences (Fig. 2).

An Illumina sequencing test was also used to supplement the qPCR data. Since sequencing library preparation requires amplification, the approaches are not independent, but the number of reads for each sequence can be used to gain insight into whether some of the oligonucleotides in the library are over- or underrepresented due to G content and position. The same set of qPCR-amplified oligonucleotides (Fig. 5a; Supplementary Table 3) was used for the sequencing analysis, and barcodes were used to distinguish among them in the read data. In order to minimize biases due to the base composition of the sequences, particularly the G content, due to the somewhat lower coupling efficiency of the G phosphoramidite, all test sequences with correct barcodes were included in the analysis by setting the Levenshtein distance to zero in the analysis software. Thus, G-rich sequences, which are more likely to have deletions, are not selectively excluded. The results are shown in Fig. 5c, which shows the number of reads for each of the eight sequences for arrays synthesized with and without debranching. With debranching, the number of reads is approximately one hundred times greater than the number of reads from the library synthesized without debranching. The reads are normalized to a value of 100 for the s0 sequence (no Gs in the test sequence) synthesized with the debranching cycle. The normalization also accounts for the additional

amplification required for the library synthesized without debranching. It is important to note that the presence of Gs in the adapters and barcodes accounts for the low number of sequencing reads, even of the s0 test sequence synthesized without debranching (Fig. 5c). In the case of the eight test sequences with varying G content, the number of reads is independent of G content when debranching is used. However, in the case of synthesis without debranching, there is a clear correlation between increasing G content and fewer reads.

In conclusion, our study demonstrates that the $O^6$-phosphitylation of guanine results in widespread fragmentation of oligonucleotides during digital photolithographic synthesis of DNA microarrays and oligonucleotide pools. The extent of fragmentation increases with the number of coupling reactions as well as the guanine content. In array-based hybridization assays, the fragmentation exhibits itself as a usable, but reduced hybridization signal, and in array-derived oligonucleotide pools or libraries, the fragmentation results in a large reduction in PCR-amplifiable sequences and in fewer sequencing reads. Fragmentation can be greatly reduced or eliminated by introducing a few-second debranching step—using standard capping reagents—immediately before the oxidation reaction of the modified phosphoramidite chemistry synthetic cycle.

## Methods
### Silanization of glass slides
Schott Nexterion D 263 borosilicate microscopic slides were functionalized with a 32.5 mmol silane solution for 4 h at room temperature with gentle agitation. The solution consisted of 95:5 (v/v) ethanol/deionized water plus

**Fig. 4 | Effect of debranching step on gene expression analysis.** Gene expression arrays were synthesized without and with a 15 s debranching step. **a** Fluorescence-scanned image of a gene expression array synthesized without debranching. **b** The scatterplot of the RMA-processed gene expression data for the array is shown in (**a**). **c** Fluorescence-scanned image of a gene expression array synthesized with the debranching step. **d** The scatterplot of the RMA-processed gene expression data for the array is shown in (**c**). The image of each gene expression shows about 0.4% of the full array. Each scatterplot is based on two arrays synthesized using the same protocol (control A and control B), and each expression value is the RMA average of three unique 60mer probes per gene. All the arrays were RMA normalized with each other; therefore, the range of log$_2$ of intensity is the same for all. The mean probe intensity for the arrays synthesized with debranching is six times greater than that of the arrays synthesized without debranching.

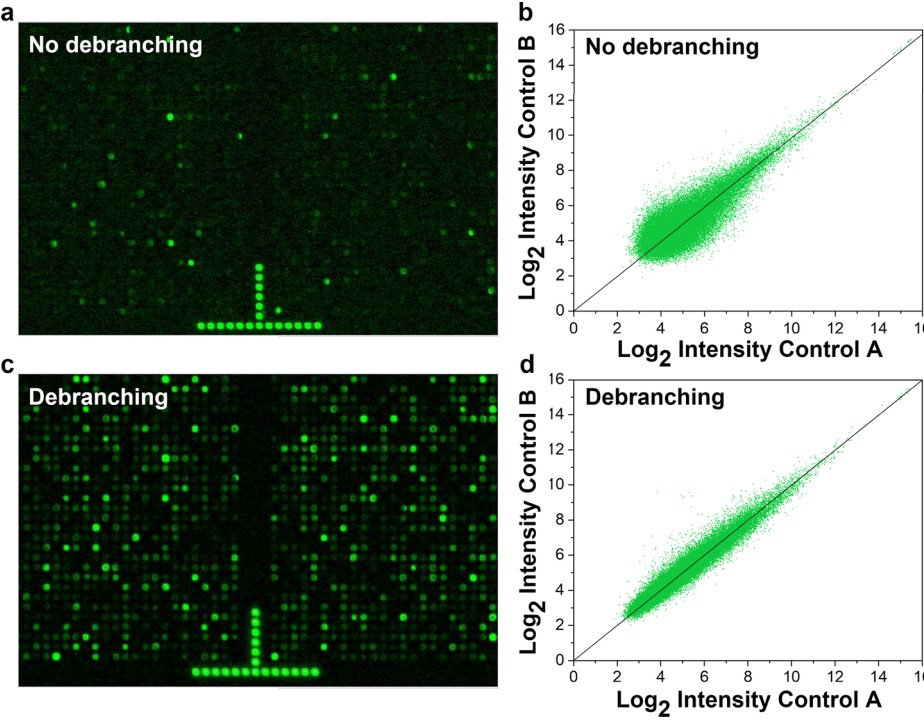

0.2% acetic acid. The glass slides were washed twice for 20 min in wash solutions containing a 95:5 (v/v) solution of ethanol/deionized water and 0.2% acetic acid, cured for 2 h at 120 °C, and cooled overnight under vacuum. The arrays used for gel analysis were functionalized with *N*-(3-triethoxysilylpropyl)-4-hydroxybutyramide (Gelest SIT8189.5) and arrays for hybridization analysis with 1,11-bis(trimethylsilyl)-4-oxa-8-azaundecan-6-ol (Gelest SIB1142.0)[63].

## Digital photolithographic synthesis of DNA microarrays and libraries

Digital photolithographic synthesis of nucleic acids uses a lightly modified form of phosphoramidite chemistry. The principle of digital photolithographic synthesis is that the deblocking step is accomplished by ultraviolet (UV) photolysis of photolabile protecting groups on the 5′-OH position of phosphoramidites (3′-position in the case of reverse synthesis[48]). Phosphoramidites with photolabile groups such as SPh-NPPOC, Bz-NPPOC, and NPPOC replace the standard DMTr phosphoramidites. 365 nm UV light from a high-power UV LED is imaged onto an array of digitally-controlled micromirrors. Micromirrors tilted into the ON position reflect light, via an optical relay, onto a glass slide within a flow cell through which solvents and reagents for phosphoramidite chemistry are introduced. This scheme is shown schematically in Fig. 6a. The position on the surface and the sequence of each oligonucleotide is determined by both the selective photodeprotection of the terminal nucleotides at each position on the surface and the order of coupling cycles, as illustrated in Fig. 6b. The optical system is shown in Fig. 6c. Light from a high-power 365 nm UV LED (Nichia NWSU333B) is collected and transmitted through a homogenizer rod, which creates a spatially uniform beam of light that can uniformly illuminate the digital micromirror device (DMD)[64]. This uniform beam of light is shaped to match the aspect ratio of the DMD and imaged onto the DMD via a set of lenses and a turning mirror. The micromirrors have two defined tilt angles, ±12° relative to the plane of the DMD, corresponding to ON and OFF. Micromirrors tilted to the OFF position reflect direct light onto an absorbing surface, while micromirrors tilted to the ON position reflect light into a unit magnification optical relay via a turning mirror. Therefore, each image projected onto the synthesis surface is a rectangular array of bright and dark squares, each corresponding to an individual micromirror. In these experiments, we used a 1080p UV DMD module (Vialux V-9501 UV) with mirror-array dimensions of 1920 × 1080, or 2,073,600 synthesis pixels, each with a dimension of about 10 × 10 μm. Extensive details on the design of the optical system, as well as on the synthesis chemistry, are available[16,20].

## DNA synthesis chemistry

Solvents and reagents are delivered to the photochemical flow cell, where synthesis takes place using an Expedite 8909 nucleic acid synthesizer. The Expedite's Luer connectors to standard columns were replaced by threaded adapters to enable connection to the flow cell with narrow inner diameter tubing (1/32″ OD, 0.4 mm ID; Vici Jour JR-T-6798). The protocol for DNA synthesis (Fig. 6d) has a 15 s coupling with 0.03 M phosphoramidites (Orgentis) activated with 0.25 M DCI (Biosolve 0004712404BS), followed by 20 s drying of the synthesis cell with helium, oxidation of the resulting phosphite triester by 0.02 M I$_2$/H$_2$O in pyridine (Sigma Aldrich L060060) for 15 s. After oxidation, photodeprotection of the photolabile groups takes place with 6 J/cm$^2$ 365 nm UV light in the presence of an exposure solvent (1% imidazole in DMSO; Sigma Aldrich 56750 or Biosolve 0001205402BS) pumped from the Expedite's auxiliary reagent port. In the case of microarrays synthesized for the purpose of hybridization assays, the exposure time was reduced to 3 J/cm$^2$. For the debranching tests and protocols, we introduce capping reagents using the Expedite's capping ports loaded with fast deprotection Cap A (tetrahydrofuran/TAC$_2$O, 100/5 (v/w); Sigma Aldrich L070000) and Cap B (*N*-methylimidazole/tetrahydrofuran/pyridine 1/8/1 (v/v/v); emp Biotech NC-0803). The debranching step is performed between the coupling and helium drying steps (Fig. 6d). Final chemical deprotection was performed by submerging the glass slide in a 1:1 solution of ethylenediamine (EDA; Sigma-Aldrich 03550) and ethanol (Sigma-Aldrich 1.07017). For the generation of DNA pools, a base-cleavable 5′-NPPOC 2′-deoxythymidine 3′-succinyl hexamide CE phosphoramidite, "base-cleavable dT" (ChemGenes CLP-2247), was coupled immediately before the start of the desired sequences using a concentration of 50 mM and a coupling time of 120 s[65]. Representative protocols for the syntheses are shown in Supplementary Table 1.

**Fig. 5 | Quantitative comparison of synthesis with and without debranching using qPCR and sequencing. a** Sequence design for the analysis. The test sequence is a 53mer polypyrimidine with a $[CT]_n$ pattern. Nucleobases in green represent the positions of G triplets substituted within the poly-pyrimidine. **b** qPCR amplification curves comparing DNA sequences synthesized with and without a 15 s debranching step. A $\Delta C_q = 7.84$ cycle shift indicates a large difference in the number of amplifiable sequences. Each curve is the average of two replicates of each sample. **c** The relative abundance of each sequence in the pool was quantified by Illumina sequencing. Normalization is relative to the s0 sequence with debranching, with the no-debranching sequences additionally scaled according to the number of additional PCR cycles needed for library preparation. Synthesis without debranching generally gives lower read counts with a significant decrease for G-rich centers closer to the 3′ end and for the G-rich sequence s7 (*, one-sided binomial test, $p < 0.01$).

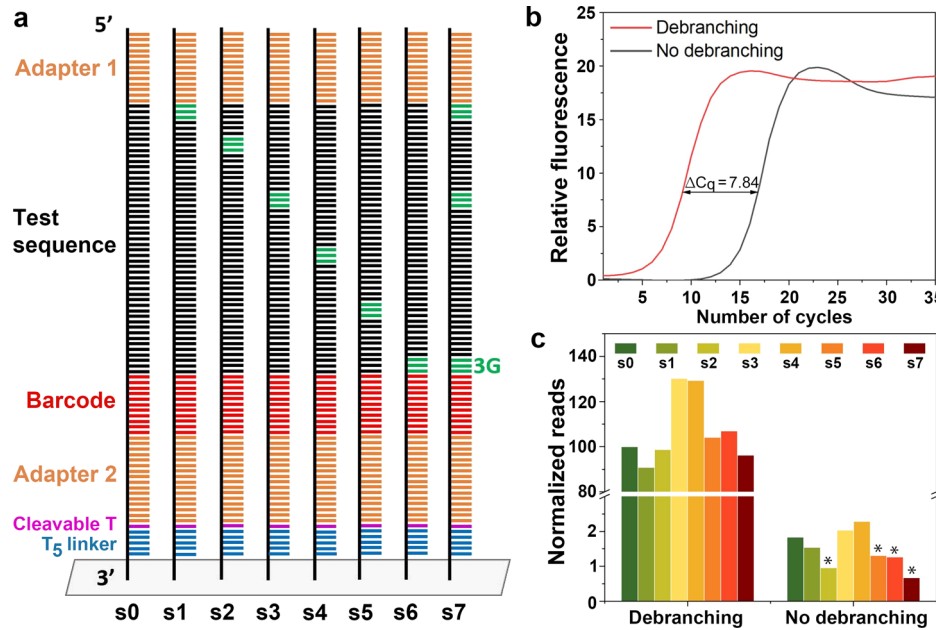

## Library elution and PAGE analysis

The arrays synthesized with base-cleavable dT at the 3′ end of the desired sequence were deprotected for 2 h in a 1:1 solution of anhydrous EDA:toluene, which cleaves the succinyl linker but leaves the DNA weakly bound to the glass surface. Deprotected slides were then washed twice with 30 mL of dry acetonitrile to remove the deprotection solution. The cleaved DNA was recovered from the surface using 200 µL of nuclease-free water. The eluate was lyophilized at 45 °C and resuspended in 30 µL of nuclease-free water. The concentration of the recovered library was determined using a Nano-Drop OneC UV–vis spectrophotometer (Thermo Fisher Scientific)[65]. The same volume of the crude sample was analyzed using 15% Novex TBE-Urea PAGE (Thermo Fisher Scientific), with GelRed staining for imaging with ChemiDoc. Three µL each of the samples were taken for gel analysis, run at 100 V for 1.5 h in 1× TBE buffer, stained with Gel Red, and referenced to a ssDNA ladder (IDT 10/60).

## Gene expression array synthesis and analysis

Gene expression microarrays had three unique probes for each of around 45,000 human mRNA transcripts, plus additional quality control probes complementary to spike-in Cy3-labeled synthetic oligonucleotides[66,67]. Two replicates of each set of slides, with and without debranching, were synthesized for hybridization with cDNA from total human mRNA. Arrays were washed twice with deionized water, dried in a microarray centrifuge, and stored in a desiccator until use. Ten microgram of human reference total RNA (Agilent) was reverse transcribed using Cy3-labeled random nonamers (Biomers) according to Ouellet et al.[68]. The gene expression arrays were hybridized in SecureSeal hybridization chambers (Grace Bio-Labs SA200) for 21 h at 42 °C with 5 µg of the Cy3-labeled cDNA in a 300 µL mixture containing 150 µL of 2× MES buffer, 3.33 µL of 10 mg/ml herring-sperm DNA (Promega D1811), 8.34 µL of acetylated BSA (Invitrogen AM2614), 11.11 µL each of "EcoBioD2_60mer", "EcoBioA1t_53mer", "QC_25mer", three Cy3-labeled synthetic spike-in controls used to verify correct synthesis and hybridization of the arrays. The arrays were washed with Non-Stringent Wash Buffer (NSWB; 6× SSPE, 0.1% Tween20) for 2 min, Stringent Wash Buffer (SWB; 100 mM MES, 0.1 M Na⁺, 0.01% Tween20) for 1 min, and Final Wash Buffer (FWB; 0.1× SSC) for a few seconds and then dried using a microarray centrifuge. All arrays were scanned with an Innopsys Innoscan 1100 scanner at 532 nm and 2 µm resolution. Fluorescence intensity data were extracted with NimbleScan

2.1 software. Expression data was normalized using Robust Multichip Analysis (RMA) and plotted for analysis[63].

## Hybridization assays

A series of 40mer pyrimidine sequences was synthesized with a $(CTCT)_n$ pattern, incorporating G-rich centers at multiple positions spanning from the 3′ to the 5′ end. Each sequence was followed by a $T_5$ linker and a 25mer sequence at the 5′ ends to enable labeling via hybridization with a 5′-Cy3 labeled complementary QC25 probe (see Supplementary Table 2 for sequence design). The arrays were hybridized for 2 h at 42 °C using the same hybridization solution as in the case of the gene expression arrays, only with 3.5 µL of 1 µM Cy3-labeled complementary DNA oligonucleotide replacing the cDNA and spike-in controls. After hybridization, the arrays were washed sequentially with NSWB (2 min), SWB (1 min), and FWB (a few seconds) solutions, then scanned with the Innopsys Innoscan at 2 µm resolution with a 0.7 PMT gain and 532 nm laser excitation. Signal intensities were extracted using NimbleScan 2.1 and subsequently analyzed in R[69].

## Quantitative PCR

Quantitative PCR (qPCR) was performed using the CFX96 Touch Real-Time PCR Detection System (Bio-Rad Laboratories) to compare the amount of amplifiable single-stranded oligonucleotides (ssDNA), synthesized with and without a debranching step. Each of the sequences with a total length of 110 nt was synthesized on a $T_5$ linker, followed by cleavable dT, Adapter 2, barcode, and Adapter 1 on the 5′ end; see Supplementary Table 3 for library design. DNA concentration was measured using the NanoDrop (see Supplementary Table 4). The qPCR reactions were performed in a total volume of 20 µL and contained 10 µL 2× KAPA SYBR FAST Universal (Roche) (without ROX), 1 µL each of 10 µM forward (0 F, ACAC-GACGCTCTTCCGATCT) and reverse primer (0 R, AGACGTGT GCTCTTCCGATCT) (Merck), 7 µL ultrapure water, and 1 µL DNA solution. Each qPCR reaction was performed in duplicate according to the following protocol. After an initial denaturation at 95 °C for 5 min, 40 cycles of a three-step PCR were performed with denaturation at 95 °C for 15 s, annealing at 54 °C for 30 s, and elongation at 72 °C for 30 s, with fluorescence measured after each elongation step. The $C_q$ values were obtained from the software Bio-Rad CFX Manager, and the actual concentration of the amplifiable sequences was calculated from the same (see Supplementary Tables 5 and 6).

**Fig. 6 | Graphical summary of digital nucleic acid photolithography. a** Light from a 365 nm LED is patterned onto the synthesis surface using a digital micromirror device. Reagents for modified phosphoramidite chemistry flow over the glass surfaces at the focal plane of the optical system. **b** Scheme for light-directed, sequence-controlled polymerization of DNA at the synthesis surface. **c** Optical system: 1—UV LED and light collection optics; 2—spatial light homogenizer; 3—illumination system aperture; 4—illumination turning mirror; 5—lens for focusing light onto the DMD; 6—1080p DMD with over 2 million digitally controlled micromirrors; 7—relay turning mirror; 8—relay front lens; 9— relay rear lens and mirror; 10—flow cell at image plane. **d** Chemical cycle for digital DNA photolithography with debranching and using 5′ Bz-NPPOC photolabile groups and DCI activator. **e** Photograph of the MAS 2.0 digital photolithographic system on an optical workstation. The fluidics delivery system, an Expedite 8909 nucleic acid synthesizer, is underneath.

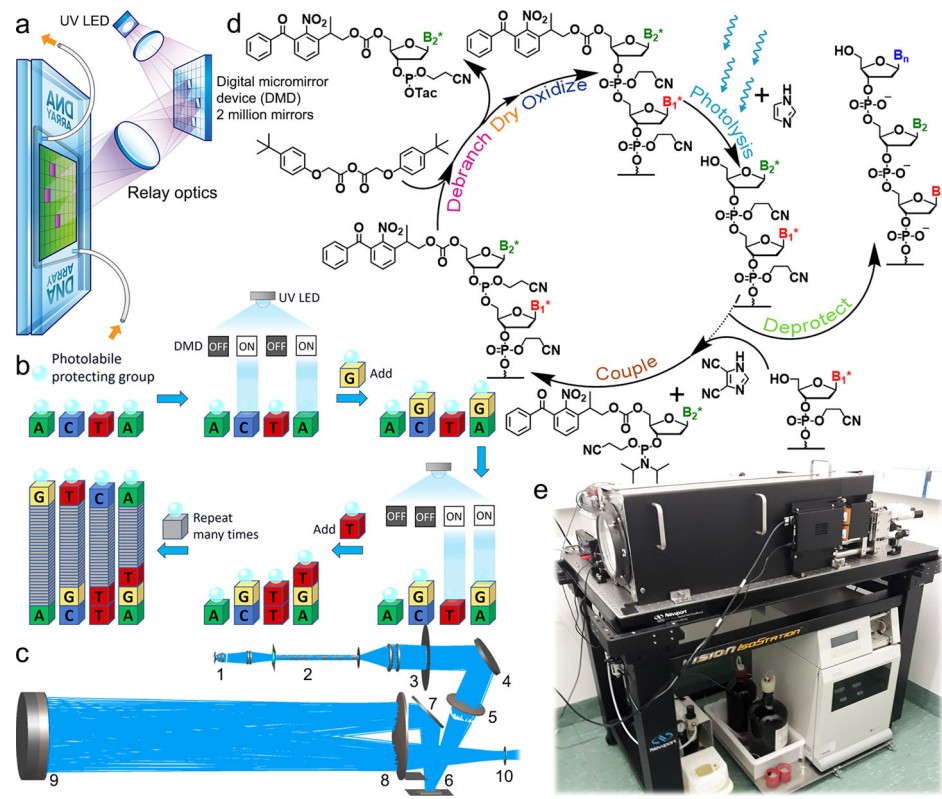

## Sequencing

For the preparation of the samples for sequencing by synthesis, the protocol of Meiser et al.[70] was followed. The same qPCR as for the quantification of DNA was conducted, but here it was stopped as soon as the fluorescence curve reached a plateau (cycle 16) to avoid overamplification of the product. The PCR product was then purified using the QIAquick PCR Purification Kit (Qiagen); the concentration was measured with Nanodrop, and the samples were diluted to 5 ng/µL. For the addition of indexed Illumina adapters, a second qPCR was performed using a similar reaction set-up as before, but with the overhang primers 2FUF, 2RIF-GM15, and 2RIF-GM17 (Supplementary Table 9) and a different protocol, denaturation at 95 °C for 15 s, annealing at 53 °C for 30 s, and elongation at 72 °C for 30 s, measuring the fluorescence after elongation, stopping after a plateau was reached at cycle 8. Ten microlitres from each PCR product was added to 10 µL of ultrapure water and run on an E-Gel EX Agarose-Gel (Thermo Fisher Scientific) with an E-Gel 50 bp DNA Ladder (Thermo Fisher Scientific). The band at 186 bp was purified using the Zymoclean Gel DNA Recovery kit (Zymo Research). The DNA concentration was measured using a Qubit 4 Fluorometer with the Qubit dsDNA HS Kit (Thermo Fisher Scientific). The samples were diluted with ultrapure water to 1 nM each and pooled together. The pooled mix was then diluted to 50 pM, and 2% PhiX Control v3 (Illumina) was spiked into the sample. The Illumina iSeq 100 i1 Reagent v2 cartridge was loaded with 20 µL of the sample, and a 150 nt paired-end sequencing run with the Illumina iSeq 100 sequencer was performed.

Sequencing data analysis was performed using a custom Python script that processed the reads through adapter trimming and barcode-based identification. First, reads were matched against the adapters ACAC-GACGCTCTTCCGATCT and AGATCGGAAGAGCACACGTCT in the 5′ to 3′ direction, which were then trimmed. Next, the trimmed reads were filtered by their length to retain reads of any length between 0 and 128 nucleotides, that is, to keep as many reads as possible. Since the barcode was designed to be 11 nucleotides long and was located in the designed positions between 0 and 10, the reads were binned by their observed barcodes in this region. Only reads with a perfect match to one of the designed barcodes were retained and binned, while all others were filtered out. This filtering ensured that only valid reads were used for the analysis, and thus, the counts of matched and assigned reads were used to measure the relative abundance of variants within the array.

## Data availability

The data underlying this article are available in the article and in its online supplementary data. Microarray gene expression data are available at ArrayExpress under accession ID E-MTAB-15163, qPCR, sequencing data, hybridization assay, and metafiles were deposited in Zenodo, available at DOI 10.5281/zenodo.15404232. The sequencing analysis script is available at Zenodo (10.5281/zenodo.17048135).

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

## Acknowledgements
The authors gratefully acknowledge financial support from the German Research Foundation (DFG) under grant 460736965 and from the European Innovation Council Pathfinder Challenge: DNA-based digital data storage under project number 101115134. The authors thank Andreas Gimpel and Robert Grass, Department of Chemistry and Applied Biosciences, ETH Zürich, for providing access to and assistance with their Illumina sequencer.

## Author contributions
S.S. carried out gel and hybridization experiments. S.S. and S.I. carried out the qPCR and sequencing experiments and analysis. O.S. and E.Y. designed and created the sequencing analysis script. S.S. and M.M.S. made all illustrations. E.Y., M.G. and J.B. supervised the project. J.B. built and programmed the MAS 2.0 device. S.S., S.I., O.S. and M.M.S. participated in the writing of the publication. All authors have read and revised the manuscript.

## Funding

## Competing interests
The authors declare no competing interests.
