## [Transparent Peer Review file · Communications Chemistry]

Efficiency of Digital Photolithographic Synthesis of Large, High-Quality DNA Libraries and Microarrays using a Guanine O6 Dephosphitylation Strategy

Corresponding Author: Professor Mark Somoza

Version 0:

Reviewer comments:

Reviewer #1

(Remarks to the Author)

The work by S. Santhosh describes chemical problems in photolithographic synthesis (branching) as well as strategies to overcome these limitations. The authors describe how they first identified the base G as problematic in synthesis, and then the effectiveness of the debranching procedure.

In general, this is an interesting study of very high quality and detail, and several techniques have been used to describe the problem and show the effectiveness of the solution.

Some comments to improve the manuscript:

- currently, the chemical effects described are only depicted in the abstract. It would be good (especially for a chemically oriented journal, if the chemical processes and hypothesis could also be displayed as a separate figure in the main manuscript.
- the authors should then add, that G-branching and debranching by capping solutions has been described in general DNA synthesis literature. The novelty of the manuscript does not lie within this discovery, but rather in the observation and quantification of the extent of these problems, as well as the identification of the minimal steps needed to remedy them in relation to photolithographic DNA synthesis. The authors may want to make this more clear in abstract and introduction. Especially, as the authors can show that the effect is quite large, and this might not have been expected from existing literature.
- in the sequencing data the authors only show the effect of the debranching step on the quantities of the oligos. They should also give qualitative data if, and to what extent, the debranching step introduces (or removes) point mutations (error rates and length distributions).
- similarly to the point above, how many sequences were filtered out during the filtering step with the very stringent selection? And does this differ between the sequences, and between branching and debranching?
- Figure 3c is very difficult to read. The authors might want to consider another way to depict the data.
- I do not understand why in Fig 5c, s0 for no-debranching is not 100, as the description writes that the data is normalized to s0.
- In case it is normalized to s0 of the debranched test sequence, the question arises why s0 with no debranching gives so much less reads (as the test sequence does not comprise any Gs)?
- Also Fig 3c, the trendlines given are very misleading, and should be removed. Firstly the main trend for both sample sets is that s3 and s4 are overrepresented (without giving an explanation), and I don't believe that the trend observed for the "no debranching" is statistically significant (or relevant).

Reviewer #2

(Remarks to the Author)

The manuscript provided ample evidence for the difference of the degree of oligonucleotide fragmentation arising from O6-phosphitylation of guanine in DNA microarray synthesis between delivering and not delivering capping reagents in each synthetic cycle. The evidence include gel electrophoresis analysis of chain cleaved/not cleaved DNA, hybridization assay, gene expression array analysis, qPCR and sequencing analysis.

The novelty of the work is not high because phosphitylation of O6-G had been well documented many decades ago, and the manuscript did not provide any new insights. However, realization of the importance of the problem in microarray DNA synthesis, and providing solutions to address the problem are significant. Therefore, this reviewer suggests either rejection and allowing resubmission after the following revisions or transferring to another journal.

Can capping using UniCap or simply additional washing after the coupling step make the same difference as the debranching method used in the manuscript? These would be better control experiments to support the theory behind the work. These reviewers suggest doing these if the manuscript will be published in Communication Chemistry. The difference does not need to be characterized using all the methods in the manuscript. One of them, for example, gel electrophoresis, is OK although more is better.

For the manuscript to be published in Communication Chemistry, these reviewers also suggest conducting the following experiment. Synthesize a sequence such as 5'-(dT)40-(dG)1-(dT)20-3' on microarray. Block the 5'-end with capping. Expose the array to repeated DNA synthesis cycles (with and without acetic anhydride capping). Deprotect and cleave. Run gel. Isolate the cleaved sequence. Analyze the bands with MS to see if the expected DNA fragments can be identified. This experiment should be able to provide more direct evidence of the theory behind the work. The suggested sequence may not be the best for the purpose. Use a different one if needed.

References 41-43 did not use acid anhydride capping, and instead the capping conditions are the same as coupling. According to the theory, on which this article is based – O6-phosphitylation of guanine causes DNA cleavage, the arrays synthesized under these conditions would be of lower quality. Whether this is true or not, it should be discussed in the manuscript either in the introduction section or in the discussion section. Interestingly, in this article *Org Biomol Chem*, 2023, 21, 9005, capping was achieved under the same conditions as coupling. An analysis of the data may provide information regarding guanine O6 phosphitylation. At a first glance, it looks like the data do not support the branching theory. If the data in the article and in references 41-43 are not sufficient, saying so in discussion is enough.

The introduction section included too much information that is not related to the problem to be addressed in this manuscript. In the meantime, information that is more relevant to the problem such as evidence of branching in the literature, proposed mechanisms, its effect on nucleotide synthesis, etc. is however not provided. Making changes regarding this is requested.

The Materials and Methods section should be focused on describing materials and procedures for the experiments. However, in its current version, a lot of unrelated information is provided. For example, "The method also allows synthesis with RNA ... and fluorescent labels (57,58)". These discussions, if not deleted, should be moved to introduction or discussion sections.

"In the case of oxidation for the previous coupling reaction (36, 40)" The writing is understandable but should be improved.

The current title may be misleading, as it suggests the manuscript presents a debranching strategy. In reality, the study compares the efficiency of synthesis with and without debranching, under varying conditions, including the presence or absence of guanine. Please consider revising the title to better reflect the comparative nature of the study.

What is the maximum oligonucleotide length that can be reliably synthesized using the digital photolithographic technique described in the manuscript? Clarification on synthesis limitations with respect to sequence length would help readers better assess the technique's practical applicability.

This review is composed by two co-reviewers, one early career reviewer and one establisher reviewer.

Reviewer #3

(Remarks to the Author)

The authors reported an approach to improving the solid-phase DNA synthesis based on the photochemical deprotection strategy. A debranching step was introduced in the synthesis cycle which effectively reduced the strand branching at the "G" site due to unwanted phosphitylation of guanine. Thorough biochemical experiments or performed to verify the positive effect of the debranching step and the results look sound. Technically it is a nice contribution to the photochemical DNA synthesis field. However, the novelty of this work in the chemistry point of view is yet not so obvious. The core of this manuscript seems to be the debranching step, while this step was rather simply introduced compared to the other standard steps (see section "introduction" line 102-103, and sub-section "DNA synthesis chemistry" line 160-161). It would help emphasizing the significance of the debranching step chemistry with more introduction and discussion throughout the manuscript before it is considered to be published in *Communications Chemistry*.

The table of concept figure (especially the two image in the middle) and Figure 1 (especially d, partly covered by e) are both a bit messy. Please consider rearranging it to improve readability.

Reviewer #4

(Remarks to the Author)

I co-reviewed this manuscript with one of the reviewers who provided the listed reports. This is part of the Communications Chemistry initiative to facilitate training in peer review and to provide appropriate recognition for Early Career Researchers who co-review manuscripts.

Version 1:

Reviewer comments:

Reviewer #2

(Remarks to the Author)

This reviewer is satisfied with the revision but would like to request a more clear description on how the new experiments in Supplementary Figure 1 were done. The description can remain in the caption in that Figure. The authors can be trusted, and there is no need for additional review concerning this particular issue.

Reply to Reviewers

Reviewer: 1

The work by S. Santhosh describes chemical problems in photolithographic synthesis (branching) as well as strategies to overcome these limitations. The authors describe how they first identified the base G as problematic in synthesis, and then the effectiveness of the debranching procedure.

In general, this is an interesting study of very high quality and detail, and several techniques have been used to describe the problem and show the effectiveness of the solution.

Author's reply: *We thank the reviewer for the positive feedback and for the positive evaluation of the quality and value of our study.*

Some comments to improve the manuscript:

- currently, the chemical effects described are only depicted in the abstract. It would be good (especially for a chemically oriented journal, if the chemical processes and hypothesis could also be displayed as a separate figure in the main manuscript.

Author's reply: *As suggested, we have added a schematic diagram showing the proposed chemical mechanism of the effect of O⁶-phosphitylation on the synthesis of DNA microarray as Fig. 2. This also replaces the optional graphical abstract.*

- the authors should then add, that G-branching and debranching by capping solutions has been described in general DNA synthesis literature. The novelty of the manuscript does not lie within this discovery, but rather in the observation and quantification of the extent of these problems, as well as the identification of the minimal steps needed to remedy them in relation to photolithographic DNA synthesis. The authors may want to make this more clear in abstract and introduction. Especially, as the authors can show that the effect is quite large, and this might not have been expected from existing literature.

Author's reply: *We revised the introduction to emphasize that we make no claim to have discovered O⁶-phosphitylation-based chain cleavage. Instead, we focus on showing how significant this problem is in photolithographic DNA microarray synthesis:*

“However, Pon et al observed that in standard solid-phase synthesis of oligonucleotides, activated phosphoramidites could phosphitylate the lactam function of previously coupled guanines (44). They showed that this reaction is followed by phosphorous group migration to N⁷, resulting in a depurination-prone N⁷-modified guanosine, and ultimately, strand cleavage during final deprotection. This chain cleavage could be mitigated by using 2-cyanoethyl or p-nitrophenylethyl protecting groups at the O⁶ position of guanine. The former requires a more complex synthesis, while the latter is a photolabile protecting group incompatible with photolithographic synthesis. A proposed alternative using only the standard protecting group for the exocyclic amine of guanine was to regenerate the intact nucleobase immediately after the coupling reaction via a nucleophilic exchange reaction with the acetate ions present in the capping reagent (45,46).

Here, we demonstrate that the digital maskless photolithographic synthesis of DNA is prone to unexpectedly extensive phosphitylation of guanine, resulting in branching followed by depurination and cleavage, likely as a result of the very large number of coupling reactions required for complex library synthesis. After quantifying the extent of these problems, we show that by adapting the results of Pon et al, this important side reaction can be effectively and efficiently eliminated using capping reagents. Unlike the acetylation reaction for capping, the

debranching reaction using the same reagents is complete within a few seconds and results in a much higher yield of intact PCR amplifiable and sequenceable DNA strands and greatly improved fluorescence signals in hybridization-based assays.”

- in the sequencing data the authors only show the effect of the debranching step on the quantities of the oligos. They should also give qualitative data if, and to what extent, the debranching step introduces (or removes) point mutations (error rates and length distributions).

Author’s reply: *Thank you for the suggestion, and we agree that this analysis would be interesting. However, the sequencing of our microarray-derived oligonucleotide pools is quite challenging due to the complex interplay between a variety of error sources present in the synthesis beyond those explicitly addressed in the manuscript, including light scattering and diffraction, which result in complex patterns of insertions, deletions, and mutations. In the context of a grant on digital data storage in DNA, we are working on developing expertise and tools to better analyze these complex sequencing datasets, and will attempt to distinguish these and explore the errors and length distribution due to debranching in this project.*

- similarly to the point above, how many sequences were filtered out during the filtering step with the very stringent selection? And does this differ between the sequences, and between branching and debranching?

Author’s reply: *We used the same filtering method for each selection and compared the results between debranched and non-debranched conditions. The main two filtering strategies we employed were to keep as many sequences as possible and keep the data pool as large as possible while also simultaneously giving the least space for misassignments of any of the sequences. This was done by disregarding deletion and insertion errors by allowing sequences for all different lengths possible (0 to 128 for an oligo length of 64). The stringent filtering was only applied to the barcodes of the sequences, allowing only 100% overlaps for the barcodes between read and design. That was important because we wanted to rule out any interference and misassignment of any read to the wrong design, so that comparison of the number of reads between the variants was possible.*

As for the comparison of the ratios of total number of filtered reads/total number of reads between the two synthesis methods shows a slightly higher rate for Debranching with 0.31 (77,404/252,932) to No Debranching with 0.22 (68,819/312,119). But these numbers do not depend on fragmentation, since fragmentation does not cause point mutations, which result in barcodes of lower synthesis quality and therefore also more sequences being filtered out. Rather, fragmentation results in strand breakage, resulting in the oligo not being amplifiable since at least one of the adapters (as shown in Fig. 6a) is missing on one of the ends. Because of this, it is also not possible to add the full Illumina adapters onto the oligo, which then results in the total loss of this oligo even before sequencing. However, due to the loss of oligos, a comparison between the ratios of different variants to each other and differences for each variant for the two syntheses is possible and makes sense. In this analysis, higher rates of oligo losses for variants containing Gs compared to s0 (without Gs) were observed for the synthesis without Debranching. Furthermore, a trend for higher oligo losses for sequences with Gs introduced earlier in the sequence was also shown. These effects are a direct result of strand breakage due to prior fragmentation. The graph and findings for this can be found in the supplementary material. In addition, the number of sequences we have obtained after filtering, and extensive information about the qPCR and sequencing information is

available in Zenodo (DOI: 10.5281/zenodo.17048135), linked in the "Data availability" section.

- Figure 3c is very difficult to read. The authors might want to consider another way to depict the data.

Author's reply: *We have redesigned Fig. 3c to make it easier to read.*

- I do not understand why in Fig 5c, s0 for no-debranching is not 100, as the description writes that the data is normalized to s0.

Author's reply: *In order to better compare the number of reads between debranching and no debranching conditions in this graph, the number of no-debranching reads was scaled down based on the number of additional PCR cycles that the sample needed during library preparation for sequencing (panel b in the same figure), so all the values are relative to s0 of the debranching library. This is mentioned in the text: "The reads are normalized to a value of 100 for the s0 sequence (no Gs in the test sequence) with debranching. The normalization also accounts for the additional amplification required for the library synthesized without debranching." Since this is an important point, the following text has been added to the figure caption: "Normalization is relative to the s0 sequence with debranching, with the no-debranching sequences additionally scaled according to the number of additional PCR cycles needed for library preparation."*

- In case it is normalized to s0 of the debranched test sequence, the question arises why s0 with no debranching gives so much less reads (as the test sequence does not comprise any Gs?)

Author's reply: *This is an important point that indeed needs to be clarified. As shown in panel a of the same figure, the actual synthesized sequences include two adapters and a barcode. These account for about half of the total length and include many Gs. Therefore, the s0 sequence of the no-debranching sequencing library is also degraded by guanine O⁶ phosphorylation, just not in the test sequence section. It might have been an option to generate G-free adapters and barcodes; however, the effect of G-free sequences on amplification and sequencing is incompletely understood. The presence of Gs in the adapters is why additional PCR cycles were needed to sufficiently amplify the no-debranching library, as mentioned above and in the manuscript. To clarify this point in the manuscript, we have added the following text to the relevant paragraph: "It is important to note that the presence of Gs in the adapters and barcodes accounts for the low number of sequencing reads, even of the s0 test sequence synthesized without debranching (Fig. 6c)."*

- Also Fig 3c, the trendlines given are very misleading, and should be removed. Firstly, the main trend for both sample sets is that s3 and s4 are overrepresented (without giving an explanation), and I don't believe that the trend observed for the "no debranching" is statistically significant (or relevant).

Author's reply: *We have removed the trendline from the graph. We believe the overrepresentation of s3 and s4 is due to PCR bias and have included this explanation in the manuscript. Nevertheless, our primary hypothesis and rationale for sequencing was to show a greater loss, and hence, fewer reads of G-rich sequences and of those with those closer to the 3' end, which are exposed to a greater number of coupling reactions (please also note that the hybridization data shown in Fig. 4c also support greater loss of G-rich sequences, particularly those with Gs closer to the 3' end). To this end, we have*

performed statistical significance testing on the no-debranching sequencing data and added this to the figure. Sequences s2, s5, s6, and s7 with no debranching show a statistically significant loss relative to the null hypothesis (one-sided binomial test, $p < 0.01$), which, in our opinion, supports the hypothesis that the loss of sequences is proportional to the exposure of Gs to subsequent coupling reactions. In addition, to allow an easier comparison between the two datasets in Figure 4c, we have added Supplementary Figure 6, which shows a side-by-side comparison.

Reviewer: 2

The manuscript provided ample evidence for the difference of the degree of oligonucleotide fragmentation arising from O⁶-phosphitylation of guanine in DNA microarray synthesis between delivering and not delivering capping reagents in each synthetic cycle. The evidence include gel electrophoresis analysis of chain cleaved/not cleaved DNA, hybridization assay, gene expression array analysis, qPCR and sequencing analysis.

The novelty of the work is not high because phosphitylation of O⁶-G had been well documented many decades ago, and the manuscript did not provide any new insights. However, realization of the importance of the problem in microarray DNA synthesis, and providing solutions to address the problem are significant. Therefore, this reviewer suggests either rejection and allowing resubmission after the following revisions or transferring to another journal.

***Author's reply:** We thank the reviewer for acknowledging the importance of our findings in the specific context of DNA microarray synthesis. While the chemistry of O⁶-phosphitylation is known, our work focuses on its quantitative impact in photolithographic DNA microarray synthesis. We believe our findings offer a simple and easy-to-apply solution to improve synthesis quality. We have revised the manuscript to reflect this more clearly.*

Can capping using UniCap or simply additional washing after the coupling step make the same difference as the debranching method used in the manuscript? These would be better control experiments to support the theory behind the work. These reviewers suggest doing these if the manuscript will be published in Communication Chemistry. The difference does not need to be characterized using all the methods in the manuscript. One of them, for example, gel electrophoresis, is OK although more is better.

For the manuscript to be published in Communication Chemistry, these reviewers also suggest conducting the following experiment. Synthesize a sequence such as 5'-(dT)₄₀-(dG)₁-(dT)₂₀-3' on microarray. Block the 5'-end with capping. Expose the array to repeated DNA synthesis cycles (with and without acetic anhydride capping). Deprotect and cleave. Run gel. Isolate the cleaved sequence. Analyze the bands with MS to see if the expected DNA fragments can be identified. This experiment should be able to provide more direct evidence of the theory behind the work. The suggested sequence may not be the best for the purpose. Use a different one if needed.

***Author's reply:** We thank the reviewers for their valuable comments. While additional washing after the coupling reaction, or the use of UniCap capping, can't reverse O⁶ phosphitylation reactions, UniCap capping in particular will change the fragmentation pattern. We performed additional library synthesis and gel experiments with the following conditions:*

1. A comparison using UniCap capping, and

2. A test sequence was synthesized with the debranching reagent and then subjected to additional synthesis cycles (without photodeprotection), both with and without debranching.

The results of these experiments are shown as Supplementary Figure 1 for a test sequence with only two guanines ($G_{20,21}$ within an otherwise 60mer T homopolymer). In our "Normal" (no debranching) cycle (Lane 2), there is extensive fragmentation as previously observed. With the "Debranching" cycle (Lane 3), minimal fragmentation is observed as previously shown in the manuscript gel images. With UniCap (Lane 4), less fragmentation is observed, but there is still a 40mer fragment visible. We attribute the cleaner gel with the use of UniCap to result from the inability of branch extension after phosphitylation with the UniCap phosphoramidite, since it is missing the 5' hydroxyl group. With UniCap, the 3' guanine depurination product (3' guanine-UniCap) would be too short to be visible in the gel. Nevertheless, the primary concern with O^6 phosphitylation is the modification of the G and possible subsequent cleavage of the full-length product, which UniCap cannot prevent. The use of extensive additional washing after coupling (Lane 5) results in no change vs the normal post-coupling wash. The use of additional synthesis cycles (without photodeprotection, equivalent to 5' capping, Lanes 6 and 7) does not show much of a change relative to the same synthesis without extra cycles. In the case of additional cycles with debranching, this is as expected. In the case of additional cycles without debranching, one might expect a slight increase (1/3, due to 60 vs 40 additional couplings post G) in the intensity of the already weak band at 40 nt. Such a change may be too subtle to see. It is also quite possible that later couplings are less likely to result in phosphitylation since the few Gs would be buried under many Ts at this stage, and less accessible to the incoming phosphoramidites. The loss of the dark fragmentation smears can be attributed to, as in the case of the use of UniCap, to the inability of any UniCap coupled to the guanine to be further extended in the absence of acid deblocking, resulting in a product too short to be visible in the gel; in this case, the product is a 3' guanine-dT. Additional analysis of these new experiments has also been added to the Supplementary Material under this new figure.

In previous works, we have performed MS analysis of the oligos synthesized on our microarray, pooling two identical arrays (~40 pmol) and using a highly sensitive mass spectrometer (Chem. Commun. 2014, 50, 12903-12906). However, the fragment products are present in far lower abundance and could not be detected by MS with our available instrumentation.

References 41-43 did not use acid anhydride capping, and instead the capping conditions are the same as coupling. According to the theory, on which this article is based – O^6 -phosphitylation of guanine causes DNA cleavage, the arrays synthesized under these conditions would be of lower quality. Whether this is true or not, it should be discussed in the manuscript either in the introduction section or in the discussion section. Interestingly, in this article, Org Biomol Chem, 2023, 21, 9005, capping was achieved under the same conditions as coupling. An analysis of the data may provide information regarding guanine O^6 phosphitylation. At a first glance, it looks like the data do not support the branching theory. If the data in the article and in references 41-43 are not sufficient, saying so in the discussion is enough.

Author's reply: References 41-43 discuss different phosphoramidites with DMT protection groups, which are generally used for capping in the case of photolithographic synthesis. In the case of photogenerated acids, coupling-based capping, like UniCap, is preferred over acetylation-based capping. In the article Org Biomol Chem, 2023, 21, 9005, 2-cyanoethyl N,N,N',N' -tetraisopropylphosphoramidite and DCI are used for

capping instead of acetylation-based capping. Capping conditions similar to coupling can still allow O⁶-phosphitylation, similar to UniCap. Even though fragmentation may appear less severe when UniCap-like phosphoramidite is used, this is due to the lack of chain elongation on the branches. More detail is now available in the text under the new Supplementary Fig. 1.

The introduction section included too much information that is not related to the problem to be addressed in this manuscript. In the meantime, information that is more relevant to the problem such as evidence of branching in the literature, proposed mechanisms, its effect on nucleotide synthesis, etc. is however not provided. Making changes regarding this is requested.

Author's reply: *Historically, microarrays have been nearly universally associated with hybridization-based genomics applications, such as gene expression, SNP detection, and copy number variation. Hybridization-based assays are very tolerant to synthesis errors as the hybridization temperature can be lowered, and the salt content increased, to compensate. Therefore, our intention in the introduction was to justify the need for a re-optimization of the synthesis to accommodate newer applications, many of which are much less error-tolerant. We also felt it important to allocate space to introducing microarray synthesis technologies, since in our experience, very few chemists or even nucleic acid chemists are familiar with the details. In particular, they may not realize that photolithographic synthesis, in the absence of an acidic deblocking step, is immune to acid-induced depurination, which is important to bring up in the context of discussing another source of depurination. That said, and also in response to a comment by Reviewer 1, we have tried to shift the emphasis of the introduction a little more towards emphasizing O⁶-phosphitylation.*

The Materials and Methods section should be focused on describing materials and procedures for the experiments. However, in its current version, a lot of unrelated information is provided. For example, "The method also allows synthesis with RNA ... and fluorescent labels (57,58)". These discussions, if not deleted, should be moved to the introduction or discussion sections.

Author's reply: *We have moved this sentence to the introduction.*

"In the case of oxidation for the previous coupling reaction (36, 40)" The writing is understandable but should be improved.

Author's reply: *We have rewritten this sentence as follows:*

"Also, in photolithographic synthesis, the substitution of light for the standard acidic deblocking step allows for a reduced oxidation frequency, but only when DCI activator (39) is used, due to its low acidity (pKa 5.2), which is insufficiently acidic to cleave the phosphite triester formed during coupling reactions (36,40)."

The current title may be misleading, as it suggests the manuscript presents a debranching strategy. In reality, the study compares the efficiency of synthesis with and without debranching, under varying conditions, including the presence or absence of guanine. Please consider revising the title to better reflect the comparative nature of the study.

Author's reply: *This is a fairly subtle point, but we have tried to address this concern by changing the title to "Efficiency of Digital Photolithographic Synthesis of Large, High-Quality DNA Libraries and Microarrays using a Guanine O⁶ Dephosphitylation Strategy."*

What is the maximum oligonucleotide length that can be reliably synthesized using the digital photolithographic technique described in the manuscript? Clarification on synthesis limitations with respect to sequence length would help readers better assess the technique's practical applicability.

Author's reply: *In this manuscript, we have analyzed 105mers for qPCR and sequencing. We have also synthesized longer sequences in other projects. For spatial transcriptomics, we generally synthesize 120mers (5' to 3' synthesis). At the moment, we are still working on clarifying any upper limit. In the absence of acid-induced depurination, there is no well-defined upper limit; however, errors from, e.g., coupling failures, will accumulate with increasing length, which imposes a soft limit depending on the error tolerance of the particular application. In the specific case of digital data storage in DNA, one of our projects at the moment, errors can be compensated for by error-correcting codes, favoring longer oligos since the fixed cost of setting up the synthesis and the glass slide and its functionalization are minimized. To address this point, we have added this sentence to the introduction: "This absence of acid-mediated depurination suggests that photolithographic synthesis is able to achieve longer oligonucleotide lengths. Sequence lengths in this project reached 105mers, and we routinely synthesize complex arrays of 120mers, and we are exploring library synthesis of 150mers and above."*

This review is composed by two co-reviewers, one early career reviewer and one establisher reviewer.

Reviewer: 3

The authors reported an approach to improving the solid-phase DNA synthesis based on the photochemical deprotection strategy. A debranching step was introduced in the synthesis cycle which effectively reduced the strand branching at the "G" site due to unwanted phosphorylation of guanine. Thorough biochemical experiments or performed to verify the positive effect of the debranching step and the results look sound. Technically it is a nice contribution to the photochemical DNA synthesis field. However, the novelty of this work in the chemistry point of view is yet not so obvious.

Author's reply: *Thank you for your feedback. Now we have updated a schematic diagram which shows the hypothesized chemical mechanism of the O⁶-phosphitylation-based chain cleavage of DNA during chemical synthesis.*

The core of this manuscript seems to be the debranching step, while this step was rather simply introduced compared to the other standard steps (see section "introduction" line 102-103, and subsection "DNA synthesis chemistry" line 160-161). It would help emphasize the significance of the debranching step chemistry with more introduction and discussion throughout the manuscript before it is considered to be published in Communications Chemistry.

Author's reply: *Thank you for the suggestion. We have updated references to include more literature on O⁶-phosphitylated branching and the techniques that have been explored through the years to save the DNA from O⁶-phosphitylation-based branching.*

The table of concept figure (especially the two image in the middle) and Figure 1 (especially d, partly covered by e) are both a bit messy. Please consider rearranging it to improve readability.

Author's reply: *Thank you for the suggestion. We have made a simplified yet detailed schematic diagram of the O⁶-phosphitylation-based DNA fragmentation on a DNA microarray (new Fig. 2).*

Reviewer: 4

I co-reviewed this manuscript with one of the reviewers who provided the listed reports. This is part of the Communications Chemistry initiative to facilitate training in peer review and to provide appropriate recognition for Early Career Researchers who co-review manuscripts.

***Author's reply:** We sincerely thank both the main and co-reviewers for their valuable feedback, which has helped improve our manuscript.*

The authors thank the reviewers and the editor for their efforts in improving the manuscript's quality. In addition to addressing the last concern of reviewer #2, the authors have modified the manuscript layout according to the Editorial request table. In particular:

The Methods section has been moved to the end, and the figures and tables, also in the Supplementary Materials file, have been reordered to match the reorganization.

The format of the references has been modified to conform to the Journal's specifications.

The number of replicates in an experiment has been added to the caption of Figure 3.

The Figures have been removed from the manuscript and uploaded as individual files.

The policy checklist has been completed

The editorial request table has been completed.

Reply to Reviewers

Reviewer #2:

1. This reviewer is satisfied with the revision but would like to request a more clear description on how the new experiments in Supplementary Figure 4 were done. The description can remain in the caption in that Figure.

***Author's reply:** We have added the following description for the synthesis methods we used for the experiment after the first paragraph of the caption of Supplementary Figure 4– “All the sequences were synthesized on the microarray following the “no debranching/normal” or “15s debranching” protocols as described in Fig. 1 in the main manuscript and Supplementary Table 2. The same volume of the sample was taken for PAGE analysis.”*

2. The authors can be trusted, and there is no need for additional review concerning this particular issue.

***Author's reply:** Thank you for your positive feedback.*